# The Most Significant Factor Affecting Gait and Postural Balance in Patients’ Activities of Daily Living Following Corrective Surgery for Deformity of the Adult Spine

**DOI:** 10.3390/medicina58081118

**Published:** 2022-08-18

**Authors:** Tomoyoshi Sakaguchi, Masato Tanaka, Naveen Sake, Kajetan Latka, Yoshihiro Fujiwara, Shinya Arataki, Taro Yamauchi, Kazuhiko Takamatsu, Yosuke Yasuda, Masami Nakagawa, Nana Takahashi, Tomoya Kishimoto

**Affiliations:** 1Department of Central Rehabilitation, Okayama Rosai Hospital, Okayama 702-8055, Japan; 2Department of Orthopaedic Surgery, Okayama Rosai Hospital, Okayama 702-8055, Japan; 3Department of Neurosurgery, University Hospital in Opole, 45-001 Opole, Poland; 4Department of Rehabilitation, Kawasaki University of Medical Welfare, Okayama 701-0193, Japan

**Keywords:** deformity of the adult spine, deformity corrective surgery, gait, postural adaptation, rehabilitation

## Abstract

*Background and Objectives*: Gait ability and spinal postural balance affect ADL in patients who underwent adult spinal deformity (ASD) surgery. However, it is still unclear how to determine what the cause is. This study was done to investigate various factors affecting gait, postural balance and activities of daily living (ADL) in patients who were operated on for ASD over a period of one year, following corrective surgery. *Materials and Method*: A cohort of 42 (2 men, 40 women, mean age, 71.1 years) who were operated on for ASD were included in this study. According to Oswestry Disability Index (ODI), based on their ADL, patients were segregated into satisfied and unsatisfied groups. Gait and postural balance abilities were evaluated before and after the operative procedure. Radiographs of spine and pelvis as well as the rehabilitation data (static balance, standing on single-leg; dynamic postural adaptation, timed up and go test (TUG); Gait Capability, walk velocity for a distance of 10 m) were acquired 12 months after surgery and analyzed. Spinopelvic parameters such as (lumbar lordosis (LL), pelvic tilt (PT), sagittal vertical axis (SVA), pelvic incidence (PI)) were marked and noted. The factors which affect patients’ satisfaction with their ADL were evaluated. *Results*: The ADL satisfied group included 18 patients (1 man, 17 women, mean age 68.6 years) and the unsatisfied group included 24 patients (1 man, 23 women, mean age 73.1 years). One year after the surgery, the two groups were tested. TUG (8.5 s vs. 12.8 s), 10 m walk velocity (1.26 m/s vs. 1.01 m/s), and single leg standing test (25 s vs. 12.8 s) were regarded as notably different. According to logistic regression analysis, only TUG was extracted as a significant factor. The cut-off value was 9.7 s, with sensitivity 75%, specificity 83%, area under the curve 0.824, and a 95% confidence interval of 0.695–0.953. *Conclusions*: A significant factor among all evaluations in postoperative ASD patients was TUG, for which the cut-off value for ADL satisfaction was 9.7 s.

## 1. Introduction

More than 30% of the total population are affected by adult spinal deformity (ASD) [1,2]. ASD is mal-alignment of the spine, the clinical picture of which is severe low back pain, neurological dysfunction, reflux esophagitis, cosmetic and mental disorders [3,4]. Many publications show the effectiveness of surgical intervention over conservative management in severe cases [5,6]. Following the surgical procedure, the elderly patients find it immensely difficult to adapt to the new posture [7]. Physical therapy is of utmost importance to regain normal activities of daily living (ADL) after ASD corrective surgery, so the patients can avoid accidental falls due to spinal imbalance [8].

A systematic review concluded that complex physical therapy following deformity correction surgery reduces fear avoidance behavior and showed better functional outcomes [9], and the rehabilitation for ASD resulted in significant postural and activity developments [10]. In recent advancements of gait analysis, a timed up and go test (TUG) was included, and has shown promising results as an analysis tool to evaluate patients’ dynamic balance [11,12]. Gait and postural balance affect ADL in patients who underwent corrective surgery [13]. Various tests were utilized to assess the ADL including velocity noted in a 10 m walk for the local community activity [14], the test of standing on a single leg to estimate the risk of an accidental fall [15], and TUG which appraised the functional impairment [16]. However, the concept of determining the most significant factor affecting gait and postural adaptation is still obscure to gauge the ADL in the patients who have undergone spinal deformity corrective surgery. The primary intent of the study is to ascertain which among the physical tests performed best predicts a remarkably better clinical outcome in patients who suffer with adult lumbar spine deformity, one year following corrective surgery.

## 2. Materials and Methods

The timed analysis has been acceded to by the review board of our establishment (No. 351). The ODI questionnaires and informed consents were duly obtained from all patients involved in this research.

### 2.1. Patient Selection

From June 2017 to May 2021, 42 consecutive patients who underwent corrective surgery at our hospital for ASD as their treatment were included in the study (Table 1). The average duration of postoperative follow-up was 361 days, a varied range of time spanning from 298 to 436 days. The inclusion criteria were age over 50 years or older and who showed radiographical evidence of at least one among the following: sagittal vertical axis (SVA) of 95 mm or more, pelvic tilt (PT) of 30 degrees or greater, and/or coronal Cobb angle of 30 degrees or greater [17]. Exclusion criteria were spinal deformities caused by neuromuscular disease, anatomically destructive infection or malignancy. Deformity correction was performed as two stages of surgery; primarily, OLIF from L1 to L5 or S1, secondary, posterior trans-pedicular fixation. The derived patient acceptable symptom state (PASS), which is defined as the highest threshold of symptom beyond which patients consider themselves well, cut-off scores could assist in making decisions as to what type of treatment could be suitable for the patients, indicating how a patient exceeds the cut-off value for “acceptable symptoms” at presentation [18]. Two groups were made based on the Oswestry Disability Index (ODI), an ADL satisfied group and ADL unsatisfied group, based on the cut-off of 18% [18]. The cohort was divided amongst the two.

### 2.2. Measurement Outcomes

#### 2.2.1. Gait Analysis (10 m Walk Test)

Under close supervision, patients were made to walk the 10 m test to assess their gait and momentum of comfortable pace of walking. This test is a frequently performed tool to assess walking speed [19]. Uncalculated distance was provided at the beginning and the termination of the walkway, which allowed the participants adequate time and space to accelerate or decelerate outside the data collection zone. This stretch of the walkway helped reduce variability during various phases of gait [20]. The average walking velocity in the respective age groups are as follows 1.24–1.34 m/s (60–69 years old), 1.13–1.26 m/s (70–79 years old), and 0.94–0.97 m/s (80–99 years old) [21]. This test has been reported to have excellent reliability [22].

#### 2.2.2. Static Balance Test (Standing on a Single Leg Test)

While being monitored and closely watched, patients were directed to stand while balancing themselves on either of their lower limbs, facing front and straight, both arms along the body. They were wearing comfortable shoes. The test was performed twice and an average of the two was considered for the data. The average results for the static balance test time according to the age were 19 s (>70 years), 10 s (70–75 years old), and 6 s (>75 years old) [23]. Single leg stand test is reported as reliable test in the systematic review [15].

#### 2.2.3. Dynamic Balance Test (Timed Up and Go Test; TUG)

TUG is measured as the time taken by a patient sitting in a chair to change position and stand up, then to walk a distance of 3 m, turn around, walk back to the chair, and sit down. TUG can be performed in various clinical physical examinations, making it a diverse tool to study activity and function. This test is a good choice to evaluate an activity-based outcome [24]. The exceptional dependability of TUG as a clinical tool, and its validity, has been validated in a systematic review [25]. The clinical assessment data were acquired 12 months following the surgery.

#### 2.2.4. Patient Reported Outcome Measures (PROM)

The Japanese version of the ODI, a lumbar spine-specific ADL evaluation method, was used [26]. According to ODI, patients were divided into an ADL satisfied group (more than 18.0%) and unsatisfied group (less than 18.0%) according to the previous report [18].

#### 2.2.5. Radiographic Measurements

The radiographic parameters (lumbar lordosis (LL), pelvic tilt (PT), sagittal vertical axis (SVA), and pelvic incidence (PI)) were measured prior to the surgery and after 12 months follow up (Figure 1). Each parameter was evaluated as recovery value, which was calculated as the difference between the postoperative and preoperative values.

### 2.3. Statistical Analysis

All the data were scrutinized for normality by the Shapiro–Wilk test. Student t-test and Mann–Whitney’s U test were used for comparison between the two groups. Logistic regression analysis with stepwise variable selection using Akaike’s Information Criterion was executed with ODI as the contingent variable and the date that showed significant differences in comparison between the two groups as independent variables.

Receiver operating characteristic curve (ROC curve) was used to examine cut-off values, sensitivity, specificity, and area under the curve for the data that showed significant differences by logistic regression analysis. The software utilized to process the data was EZR [27] and *p* < 0.05 was noted as remarkably significant. All numerical values of the cohort expressed are as mean ± standard deviation (SD).

## 3. Results

As for the preferred level of fusion, all LIV were pelvis and UIV were T4:1, T6:1, T9:3 and T10:37. Preoperative and one-year-follow-up ODI of these patients were 42.8 ± 13.1 and 27.3 ± 19.4, respectively (*p* = 0.00074). The satisfied group was 18 patients (1 man, 17 women, mean age 68.6 years) and the unsatisfied group was 24 patients (1 man, 23 women, mean age 73.1 years). Spinopelvic parameters of the two groups are summarized in Table 2. All spinopelvic parameters such as post op SVA, PT and PI-LL showed no statistically significant variance between the two groups. Results of both gait and postural adaptation tests performed by the two groups are listed in Table 3. The data clearly shows significant improvement in 10 m walk velocity and TUG in the postoperative period.

The ODI satisfied group were relatively younger, but no difference of gender distribution and BMI was seen, compared with the ODI unsatisfied group (Table 4). One year after the surgery, 10 m walk velocity (1.26 m/s vs. 1.01 m/s), timed up and go (TUG) test (8.5 s vs. 12.8 s) and single leg standing test (25 s vs. 12.8 s) were regarded as significantly different between the two groups. According to logistic regression analysis, only TUG was extracted as an important value. The cut-off value was 9.7 s, with sensitivity 75%, specificity 83%, area under the curve 0.824, and a 95% confidence interval 0.695–0.953 (Table 5 and Figure 2).

## 4. Discussion

A large number of excellent results were reported about corrective ASD surgery with regards to the ODI questionnaire [28,29]. Furthermore, several reports indicated that the long spinal fixation did not affect patients’ postoperative ADL satisfaction [30,31]. However, there is no report investigating the factors of gait ability and spinal balance which affect patients’ ADL by ODI. The 10 m walk velocity was reported to be a remarkable factor to join the local community activity [14]. The single leg stand test can be evaluated the risk of an accidental fall [15]. In this study, these two factors did not significantly affect the patients’ ADL satisfaction. Only TUG was graded as an important factor according to logistic regression analysis. TUG includes the combination of gait velocity, along with static and dynamic balance, which are also related to patients’ ADL with the Barthel index [24]. Watanabe et al. reported TUG was related to health-related quality of life (Scoliosis Research Score-22) at one year after surgery [32], so our results were compatible with their report. TUG was also noted as a prominent tool in the assessment of patients with lumbar canal stenosis in their objective functional impairment [16].

Using TUG of 9.7s as a cut-off value had 75% sensitivity and 82% specificity in the present study, which means that it intermediately predicts [33] the value of patients’ ADL satisfaction at one year after corrective ASD surgery. Other reports presented TUG cut-off value as 11.0 s to musculoskeletal ambulation disability symptom complex [34], and at 13.6 s to accidental fall risk [35]. A TUG of 9.7 s as cut-off value for ADL is relatively small, because two other studies were done based on critical and risk assessment conditions of the patients. The main evaluation points of TUG were forward tilting motion when standing up and sitting down, and spinal balance when turning around. Postoperatively, for ASD patients who underwent long spinal fusion, segmental motion was restricted. Hence, the patients require orientation and rehabilitation to maintain their spinal balance. The ODI questionnaire includes activities which require spinal dynamic balance, such as lifting, social life and travelling, and their ease of performance.

There are several reports about TUG in patients with ASD [12,36]. As the sagittal correction was large, TUG became shorter than the preoperative value [36]. In our report done previously, we noted that recovery of TUG needs six months after surgery [12]. The study included normal elderly people, and the effect of core instability and strength training. The results revealed the importance of trunk muscle exercise, and that a stable trunk made the movement of the extremities easier, enabling a smooth function, thereby improving TUG time [37]. However, the trunk muscles were reported to be less powerful than normal elderly people [38] in patients recovering from ASD surgery, for whom the dynamic balance rehabilitation is very important, especially to stand-up, turn around and have a smooth gait. For clinical use of our results, if TUG of ASD patient is more than 9.7 s, it is very likely the patient feels unsatisfied in their daily living. This patient should be recommended rehabilitation such as a trunk muscle exercise to stabilize dynamic body balance and a quadriceps femoris exercise which is important to standing up (these two muscle groups are key to improve TUG).

For this study we noted certain limitations as follows. The cohort was numerically small and the follow-up time was relatively short. The gender distribution of this study was unequal. We did not perform any evaluation of muscle power nor center of gravity sway test. Patient related outcome measures (PROMs) other than ODI were not used in this study.

## 5. Conclusions

The most significant factor to evaluate ADL for postoperative patients with ASD was TUG. The cut-off value for ADL satisfaction for the ASD patients was 9.7 s at the end of one year after surgery.

## Figures and Tables

**Figure 1 medicina-58-01118-f001:**
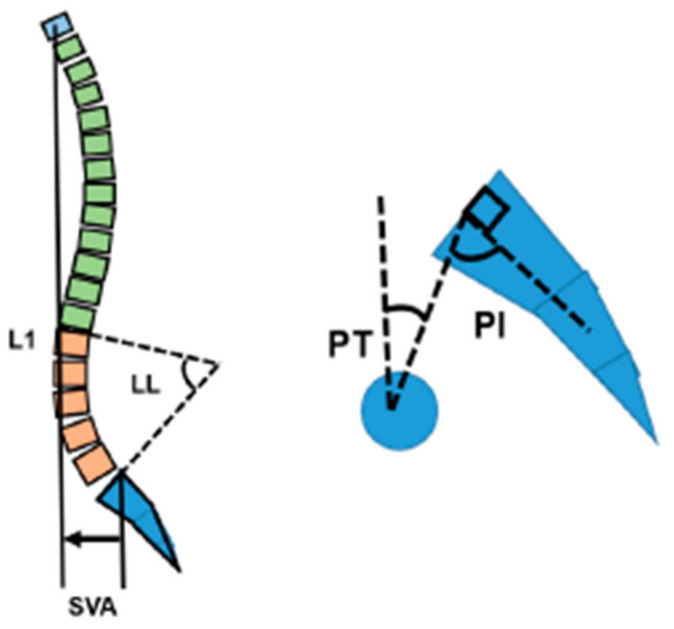
Spinopelvic parameter, SVA; sagittal vertical axis, LL; lumbar lordosis, PT; pelvic tilt, PI; pelvic incidence.

**Figure 2 medicina-58-01118-f002:**
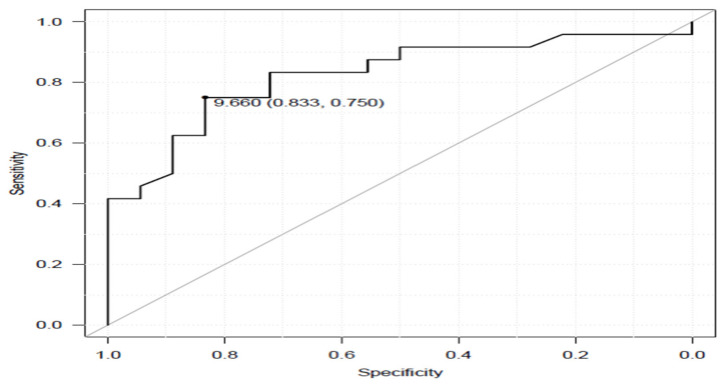
Roc curve of tug for ODI satisfied group and ODI unsatisfied group.

**Table 1 medicina-58-01118-t001:** Patient demographic.

Patient	*n* = 42
Age at surgery (year)	71.1 ± 7.6
Height (cm)	151 ± 6.4
Weight (kg)	51.9 ± 8.2
Bone mineral index (kg/m^2^)	22.8 ± 3.6

**Table 2 medicina-58-01118-t002:** Spinopelvic parameters of the two groups.

	Satisfied (*n* = 18)	Unsatisfied (*n* = 24)	*p*-Value
Preoperative SVA	102.6 ± 42.7	103.4 ± 48.4	0.74
Postoperative SVA	42.7 ± 22.6	39.3 ± 31.2	0.42
Preoperative PI	56.2 ± 8.3	55.5 ± 11.8	0.31
Postoperative PI	58.5 ± 8.7	58.0 ± 11.8	0.95
Preoperative LL	16.6 ± 16.9	7.9 ± 13.3	0.11
Postoperative LL	49.5 ± 10.3	42.5 ± 9.7	0.05
Preoperative PI-LL	39.5 ± 18.6	44.6 ± 15.3	0.33
Postoperative PI-LL	6.5 ± 13.3	10.0 ± 17.3	0.58
Preoperative PT	36.8 ± 9.7	33.2 ± 9.1	0.47
Postoperative PT	18.5 ± 9.3	19.5 ± 10.7	0.63

SVA: sagittal vertical axis, PI: pelvic incidence, LL: lumbar lordosis, PT: pelvic tilt.

**Table 3 medicina-58-01118-t003:** Results of gait and posture balance test.

	Preoperative Data	Postoperative Data	*p*-Value
10m walk velocity (m/s)	1.01 ± 0.3	1.11 ± 0.3	*p* < 0.01
Timed up and go test (s)	12.8 ± 5.9	11.0 ± 4.8	*p* < 0.01
Single leg stand test (s)	16.0 ± 12	17.4 ± 12	0.65

**Table 4 medicina-58-01118-t004:** Parameters of the two groups.

	Satisfied (*n* = 18)	Unsatisfied (*n* = 24)	*p*-Value	Power Analysis
Age at surgery (year)	68.6 ± 8.8	73.1 ± 5.8	0.047	0.58
Gender (male/female)	1/16	1/23	0.836	0.05
BMI (kg/m^2^)	23.2 ± 3.2	22.4 ± 3.9	0.234	0.16
10m walk velocity (m/s)	1.26 ± 0.2	1.01 ± 0.3	*p* < 0.01	0.76
Timed up and go test (s)	8.5 ± 1.3	12.8 ± 5.6	*p* < 0.01	0.82
Single leg stand test (s)	25.0 ± 8.9	11.4 ± 10.4	*p* < 0.01	0.99
Visual analog scale (mm)	17.0 ± 2.1	20.1 ± 1.3	0.416	0.08
SVA RV (mm)	60.7 ± 35.8	64.0 ± 40.8	0.747	0.05
PI RV (degree)	2.3 ± 3.8	6.0 ± 8.5	0.116	0.41
PT RV (degree)	17.7 ± 8.7	13.8 ± 11.6	0.227	0.21
LL RV (degree)	32.8 ± 16.2	34.6 ± 16.6	0.748	0.05

SVA: sagittal vertical axis, LL: lumbar lordosis, PT: pelvic tilt, PI: pelvic incidence, RV: recovery value.

**Table 5 medicina-58-01118-t005:** Results of logistic regression analysis.

	Odds Ratio	95% Confidence Interval	*p*-Value
Age at surgery	0.98	0.87–1.12	0.84
10 m walk velocity	247	0.29–210,561	0.12
Timed up and go	3.14	1.05–9.38	0.03
Single leg stand test	0.92	0.85–1.01	0.07

## Data Availability

The data presented in this study are available in the article.

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
