# Peer review of "The Most Significant Factor Affecting Gait and Postural Balance in Patients’ Activities of Daily Living Following Corrective Surgery for Deformity of the Adult Spine"

_medicina, 2022, doi:10.3390/medicina58081118_

Round 1

Reviewer 1 Report

Introduction

"Following the surgical procedure, the elderly patients find it immensely difficult to adapt to the new posture". Can you provide any reference to support this statement?

"Gait and postural balance affect ADL in patients who underwent corrective surgery." Again, can you provide any reference to support this statement?.

"However, it is still unclear how to evaluate what is the most significant factor and its cut-off value. ". I don't quite understand the sentence. What is the most significant factor of what?. Its cut-off value? So, we are talking about a measurement tool, not a factor. Please. Rewrite this sentence. The research objectives are not at all clear.

Patient selection.

"42 patients who underwent corrective surgery at our hospital for ASD, as their treatment, were included in the study". "Two groups were made, based on the Oswestry Disability Index (ODI), ADL satisfied group and ADL unsatisfied group, based on the cut-off of 18%"

As far as I understand, the groups were created according to the ODI at the time of follow-up. What is the follow up time (mean and range)?. Usually, only patients with at least 1 year of follow-up should be included.

It is surprising that such a heterogeneous group of patients (sagittal deformity, coronal deformity or a mixture of the two) were treated with the same type of surgery and the same levels of fixation. Could they justify this procedure?.

Do you have any data on the clinical status of the patients before surgery?. Do you have any data on the clinical change (of ODI, for example) with surgery?.

Certainly, the limitations of the study are far more than those noted in the discussion. This section should be revised.

In general terms, the information provided in the paper is interesting. However, numerous modifications will have to be made.

The main objective of the study is to determine which physical performance test best predicts the clinical status of adult spine deformity patients after surgery. Information about these tests should be provided in the introduction. It would also be interesting for the reader if you could provide some information about PASS. A clear explanation of the study's objectives.

Author Response

Dear respective reviewer 1,

First of all, we appreciate your enormous efforts and contribution for our paper.

Introduction

"Following the surgical procedure, the elderly patients find it immensely difficult to adapt to the new posture". Can you provide any reference to support this statement?

Thank you for your important comment. We added the reference according to your advice.

  1. Yagi M, Ohne H, Kaneko S, Machida M, Yato Y, Asazuma T. Does corrective spine surgery improve the standing balance in patients with adult spinal deformity? Spine J. 2018 Jan;18(1):36-43. doi: 10.1016/j.spinee.2017.05.023. Epub 2017 May 23. PMID: 28549902.

"Gait and postural balance affect ADL in patients who underwent corrective surgery." Again, can you provide any reference to support this statement?

We appreciate your comment. We added the reference according to your advice.

  1. Makino T, Takenaka S, Sakai Y, Yoshikawa H, Kaito T. Factors related to length of hospital stay after two-stage corrective surgery for adult spinal deformity in elderly Japanese. J Orthop Sci. 2021 Jan;26(1):123-127. doi: 10.1016/j.jos.2020.02.016. Epub 2020 Mar 24. PMID: 32220467.

"However, it is still unclear how to evaluate what is the most significant factor and its cut-off value. ". I don't quite understand the sentence. What is the most significant factor of what?. Its cut-off value? So, we are talking about a measurement tool, not a factor. Please. Rewrite this sentence. The research objectives are not at all clear.

Thank you for your valuable comment. We changed the sentence as follows.

However, it is still unclear what is the most significant factor of gait and postural balance to evaluate ADL of the patients after spinal corrective surgery.

Patient selection.

"42 patients who underwent corrective surgery at our hospital for ASD, as their treatment, were included in the study". "Two groups were made, based on the Oswestry Disability Index (ODI), ADL satisfied group and ADL unsatisfied group, based on the cut-off of 18%"

As far as I understand, the groups were created according to the ODI at the time of follow-up. What is the follow up time (mean and range)?. Usually, only patients with at least 1 year of follow-up should be included.

We appreciate your comment. We added the sentences as follows.

The average follow-up time was 361 days, ranging 298 to 436 days.

It is surprising that such a heterogeneous group of patients (sagittal deformity, coronal deformity or a mixture of the two) were treated with the same type of surgery and the same levels of fixation. Could they justify this procedure?.

Thank you for your important comment.

That plan is our basic strategy for ASD correction surgery.

For spinal corrective surgery, we usually choose pelvis as LIV because we need strong LIV anchor for osteoporotic spine. UIV of these patients were a little different; T4:1,T6:1,Th9:3, T10:37. So we corrected the sentence.

Do you have any data on the clinical status of the patients before surgery?. Do you have any data on the clinical change (of ODI, for example) with surgery?.

We appreciate your important comment. We added the sentence as follows.

Preoperative and one-year-follow-up ODI of these patients were 42.8±13.1 and 27.3±19.4, respectively (p= 0.00074).

Certainly, the limitations of the study are far more than those noted in the discussion. This section should be revised.

Thank you for your valuable advice. We changed the limitations of the study as follows.

For this study we noted certain limitations which were as follows, the cohort was numerically small and the follow-up time was relatively short. The gender distribution of this study was unequal. We did not perform any evaluation of muscle power nor center of gravity sway test. Other PROs but ODI were not used in this study.

In general terms, the information provided in the paper is interesting. However, numerous modifications will have to be made.

The main objective of the study is to determine which physical performance test best predicts the clinical status of adult spine deformity patients after surgery. Information about these tests should be provided in the introduction.

We appreciate your thoughtful suggestion.

Several tests were used to evaluate ADL such as 10 m walk velocity for the local community activity [28], the single leg stand test for the risk of an accidental fall [19], and TUG for functional impairment [30].

It would also be interesting for the reader if you could provide some information about PASS.

Thank you for your comment. We added the sentence as follows.

Patient acceptable symptom state (PASS) has been defined as the highest level of symptom beyond which patients consider themselves well. Recently, PASS has been reported as important evaluation for spinal surgery.

van Hooff ML, Mannion AF, Staub LP, Ostelo RW, Fairbank JC. Determination of the Oswestry Disability Index score equivalent to a "satisfactory symptom state" in patients undergoing surgery for degenerative disorders of the lumbar spine-a Spine Tango registry-based study. Spine J. 2016 Oct;16(10):1221-1230. doi: 10.1016/j.spinee.2016.06.010. Epub 2016 Jun 22. PMID: 27343730.

A clear explanation of the study's objectives.

We appreciate your important suggestion. According to your advice, we changed the sentence as follows.

The main objective of the study is to determine which physical performance test best predicts the clinical status of adult spine deformity patients after surgery.

Reviewer 2 Report

Dear authors,

thank you for the submission of the manuscript. 

Please clarify:

1) Were 42 consecutive patients included in the study?

2) Why was always the identical procedure done? Please provide exact spinopelvic parameters before and after surgery?

3) Was the gait pattern analyzed prior to the operation? If yes, please provide  the information. If no, please explain why.

4) It is stated that the unsatisfied group was almost 5 years older than the satisfied group. Please explain and argue if this could have affected the gait pattern.

5) Please provide a statistical power-analysis. 

Author Response

Dear respective reviewer 2,

First of all, we appreciate your enormous efforts and contribution for our paper.

  • Were 42 consecutive patients included in the study?

Thank you for your comment.

YES. We added this word.

2) Why was always the identical procedure done? Please provide exact

spinopelvic parameters before and after surgery?

UIVs were different, so we added the sentence as follows.

For the fusion level, all LIV were pelvis and UIV were T4:1, T6:1, T9:3, T10:37.

And we added spinopelvic parameters.

3) Was the gait pattern analyzed prior to the operation? If yes, please

provide  the information. If no, please explain why.

We appreciate your comment.

We added the data as follows.

4) It is stated that the unsatisfied group was almost 5 years older than

the satisfied group. Please explain and argue if this could have

affected the gait pattern.

Thank you for your comment.

Gait and posture balance must be affected by patients’ age. However, according to our logistic regression analysis, the patients’ age (5 years difference) was not extracted as an important value in this study,.

5) Please provide a statistical power-analysis.

We appreciate your important comment. We added the statistical power-analysis.

Reviewer 3 Report

The authors provide an interesting insight regarding the outcome of patients treated for spinal deformities. The manuscript is well written and structured and the analysis is accurate, underlining hot aspects of the post-operative impact of such surgery. Some minor concerns regard the presence of some results in the first paragraphs of method section. I would invite the authors to relocate the reported data in the more appropriate results section. Moreover the authors should improve the discussion proposing how their results could influence the daily clinical practice and in this sense what kind of strategies or methods could be adopted to reduce the unsatisfied patients according to their data. 

Author Response

Dear respective reviewer 3,

First of all, we appreciate your enormous efforts and contribution for our paper.

Comments and Suggestions for Authors

The authors provide an interesting insight regarding the outcome of patients treated for spinal deformities. The manuscript is well written and structured and the analysis is accurate, underlining hot aspects of the post-operative impact of such surgery. Some minor concerns regard the presence of some results in the first paragraphs of method section. I would invite the authors to relocate the reported data in the more appropriate results section.

Thank you for your important comments.

We relocated the data according to your advice.

Moreover the authors should improve the discussion proposing how their results could influence the daily clinical practice and in this sense what kind of strategies or methods could be adopted to reduce the unsatisfied patients according to their data.

We appreciate your valuable comment.

For clinical use of our results, if TUG of ASD patient is more than 9.7s, it is very likely the patient feel unsatisfied their daily living. This patient should be recommended rehabilitation such as trunk muscle exercise to stabilize dynamic body balance and quadriceps femoris exercise which is important to standing up (these two muscle groups are key to improve TUG).

Round 2

Reviewer 1 Report

The paper has been significantly improved according to the recommendations. In my opinion it deserves to be published.